# ASCOT identifies key regulators of neuronal subtype-specific splicing

Jonathan P. Ling [1,2], Christopher Wilks[3,4], Rone Charles[3,4], Patrick J. Leavey [2], Devlina Ghosh[2], Lizhi Jiang[2], Clayton P. Santiago[2], Bo Pang[2], Anand Venkataraman [2], Brian S. Clark [5], Abhinav Nellore[6,7,8], Ben Langmead [1,3,4,13]* & Seth Blackshaw [1,2,9,10,11,12,13]*

Public archives of next-generation sequencing data are growing exponentially, but the difficulty of marshaling this data has led to its underutilization by scientists. Here, we present ASCOT, a resource that uses annotation-free methods to rapidly analyze and visualize splice variants across tens of thousands of bulk and single-cell data sets in the public archive. To demonstrate the utility of ASCOT, we identify novel cell type-specific alternative exons across the nervous system and leverage ENCODE and GTEx data sets to study the unique splicing of photoreceptors. We find that *PTBP1* knockdown and *MSI1* and *PCBP2* overexpression are sufficient to activate many photoreceptor-specific exons in HepG2 liver cancer cells. This work demonstrates how large-scale analysis of public RNA-Seq data sets can yield key insights into cell type-specific control of RNA splicing and underscores the importance of considering both annotated and unannotated splicing events.

[1] Kavli Neuroscience Discovery Institute, Johns Hopkins University, Baltimore, MD, USA. [2] Solomon H. Snyder Department of Neuroscience, Johns Hopkins University, Baltimore, MD, USA. [3] Department of Computer Science, Johns Hopkins University, Baltimore, MD, USA. [4] Center for Computational Biology, Johns Hopkins University, Baltimore, MD, USA. [5] John F. Hardesty, MD Department of Ophthalmology and Visual Sciences, Washington University, St. Louis, MO, USA. [6] Department of Biomedical Engineering, Oregon Health and Science University, Portland, OR, USA. [7] Department of Surgery, Oregon Health and Science University, Portland, OR, USA. [8] Computational Biology Program, Oregon Health and Science University, Portland, OR, USA. [9] Department of Ophthalmology, Johns Hopkins University School of Medicine, Baltimore, MD, USA. [10] Department of Neurology, Johns Hopkins University School of Medicine, Baltimore, MD, USA. [11] Center for Human Systems Biology, Johns Hopkins University School of Medicine, Baltimore, MD, USA. [12] Institute for Cell Engineering, Johns Hopkins University School of Medicine, Baltimore, MD, USA. [13] These authors jointly supervised this work: Ben Langmead, Seth Blackshaw. *email: langmea@cs.jhu.edu; sblack@jhmi.edu

RNA-Seq is a powerful tool for studying gene expression, alternative splicing, and post-transcriptional regulation. Its utility has made it one of the most common experimental data types stored in the Sequence Read Archive[1] and other related international archives[2]. However, public archives store raw, unprocessed data. Drawing new conclusions from many raw RNA-Seq data sets requires a level of computational power and expertise that is out of reach for most labs. Likewise, the need to analyze this data from scratch leads to unnecessary duplications of effort across research groups[3,4]. To address this, we previously developed a bioinformatics pipeline (Rail-RNA)[5,6] and created the recount2 (ref. [7]) resource and accompanying Snaptron[8] query engine. Together, these allow researchers to query publicly available RNA-Seq data in a standardized and reproducible manner. In this work we focus on the alternative splicing use case for RNA-Seq data.

Alternative splicing of pre-mRNA (RNA splicing) is a highly regulated process that generates extensive transcriptomic and proteomic diversity across all cell types. RNA splicing is governed by both *cis*-regulatory elements (specific sequences in the pre-mRNA that influence the strength of a splice site) and *trans*-acting splicing factors (RNA-binding proteins that can act as either splicing enhancers or repressors). RNA-Seq has accelerated our understanding of how alternative splicing networks are coordinated, in part through the meta-analysis of RNA-Seq data gathered from many independent experiments[9,10]. Numerous algorithms for alternative splicing analysis have been developed[11–27], including several recent studies that propose useful models for studying complex splicing patterns in RNA-Seq data[11,13,25]. However, there is a need for new methods that can summarize alternative splicing across thousands of public data sets in a unified manner, without relying on prior transcript annotation[28,29].

Our work aims to make alternative splicing analysis of public RNA-Seq data accessible to the general researcher by reducing computational barriers to entry. We have developed alternative splicing catalog of the transcriptome (ASCOT), a resource that allows users to query alternative splicing and gene expression across a wide range of cell types and tissues from mouse and human. ASCOT uses an annotation-free method to quickly identify splice-variants in large-scale databases of splice junction counts derived from the public archive[8]. ASCOT performs a rapid and computationally inexpensive "junction-walking" strategy to calculate the percent spliced-in (PSI) ratio for a given exon, whereby inclusion and exclusion junctions are predicted using only counts from a splice junction database (Supplementary Fig. 1). ASCOT focuses on identifying binary splicing decisions, as these represent the majority of alternative splicing events (Supplementary Fig. 2). Although it is possible to capture more complex splicing variation with nested decision trees, here we focus on four easily interpretable and binary splicing patterns: cassette exons, alternative splice site exon groups, linked exons, and mutually exclusive exons. This exon-centric approach can rapidly capture much of the alternative splicing in the transcriptome, while simultaneously calculating each exon's PSI across thousands of indexed data sets.

We then used ASCOT to analyze data sets from a manually curated list of purified mouse cell types (732 run accessions) in the Sequence Read Archive (SRA), tissue data sets from the human Gene Tissue Expression Consortium[30] (GTEx – 9,662 run accessions), shRNA-Seq data sets from the ENCODE Project[10,31,32] (1,159 run accessions), 43 single-cell studies (33,303 cells) in human and mouse including the Allen Brain Institute adult mouse primary visual cortex study[33], and over 50,000 other human RNA-Seq run accessions from the SRA as generated for the recount2 database. To demonstrate the utility of our work, we used ASCOT to characterize the cell type-specific splicing patterns of rod photoreceptors.

The vertebrate nervous system derives much of its transcriptomic and proteomic diversity from highly specific alternative splicing patterns that are not present elsewhere in the body[34]. Many neuronal subtypes, such as rod photoreceptors, also exhibit alternative exons that are only detected in that specific cell type[35–38]. Photoreceptors are cells within the retina that sense light and transduce this information for the brain. These sensory neurons are unique in terms of morphology, metabolism, and function — characteristics that may require specialized alternative exons[35,39–47]. Photoreceptor degeneration is the main cause of hereditary blindness in the developed world. While some forms of vision loss can be successfully managed with therapies such as angiogenesis inhibitors, prosthetic devices, or tissue transplantation, few treatments exist for blindness that is directly caused by photoreceptor degeneration. Understanding how photoreceptor-specific splicing patterns emerge may facilitate development of cell-based regenerative strategies for treating photoreceptor dystrophies.

## Results

**Identification of cell type-specific alternative exons.** We first tested if we could use ASCOT to identify neuron-specific splicing patterns (Fig. 1). Publicly available RNA-Seq data sets from mouse cell types across the body were manually curated from the SRA and incorporated into ASCOT as a data compilation called MESA: mouse expression and splicing atlas. All data is openly available at http://ascot.cs.jhu.edu/. These cell types were isolated by different research groups using fluorescence-activated cell sorting (FACS) or affinity purification. As expected, we identified many exons that were highly utilized (high PSI) in neurons but skipped by other cell types (Fig. 1a). We also identified exons exhibiting the opposite pattern, having high PSI across most cell types but low PSI in neuronal cell types. Exons enriched in neurons could be further categorized based on their use in muscles and/or pancreatic islet cells. Finally, an analysis of NRL-positive rod photoreceptors[48], profiled at several timepoints from postnatal day 2 (P2) to P28, revealed that rods utilize only a subset of pan-neuronal exons, and exclude many other exons that have high PSI across other neuronal subtypes. This is consistent with the observation that rods do not express many common neuronal splicing factors[35] (Supplementary Fig. 3). Next, we tested whether we could identify alternative exons utilized only by a single brain cell type, despite near ubiquitous expression of the associated gene. We found many examples of cell type-specific exons, of which ~70% (168/239) were entirely unannotated in GENCODE release M20 (Supplementary Data 1, RT-PCR validation in Supplementary Fig. 4). For instance, an exon in *Sptan1* is only used by cochlear hair cells (Fig. 1b), an exon in *Cnih1* is selectively used by excitatory pyramidal neurons (Fig. 1c), and an exon in *Exoc6b* is selectively used in oligodendrocytes (Fig. 1d).

**Photoreceptor-specific exons shared between mouse and human.** We next sought to cross-validate our mouse cell type results with RNA-Seq data from human tissue. The Genotype-Tissue Expression (GTEx) project is a public archive of 9,662 human RNA-Seq samples across 53 tissues, although it is missing retinal tissue. We therefore analyzed GTEx data sets, supplemented with RNA-Seq data from peripheral retina[49], and identified tissue-specific alternative exons (Supplementary Data 2). We identified ~104 exons that are selectively utilized in human retina, compared to all other GTEx human tissues (Fig. 2a, b). In the mouse genome, we identified ~88 exons that were enriched in rod photoreceptors, compared to all other mouse cell types in MESA. Cross referencing human retina-specific exons and mouse

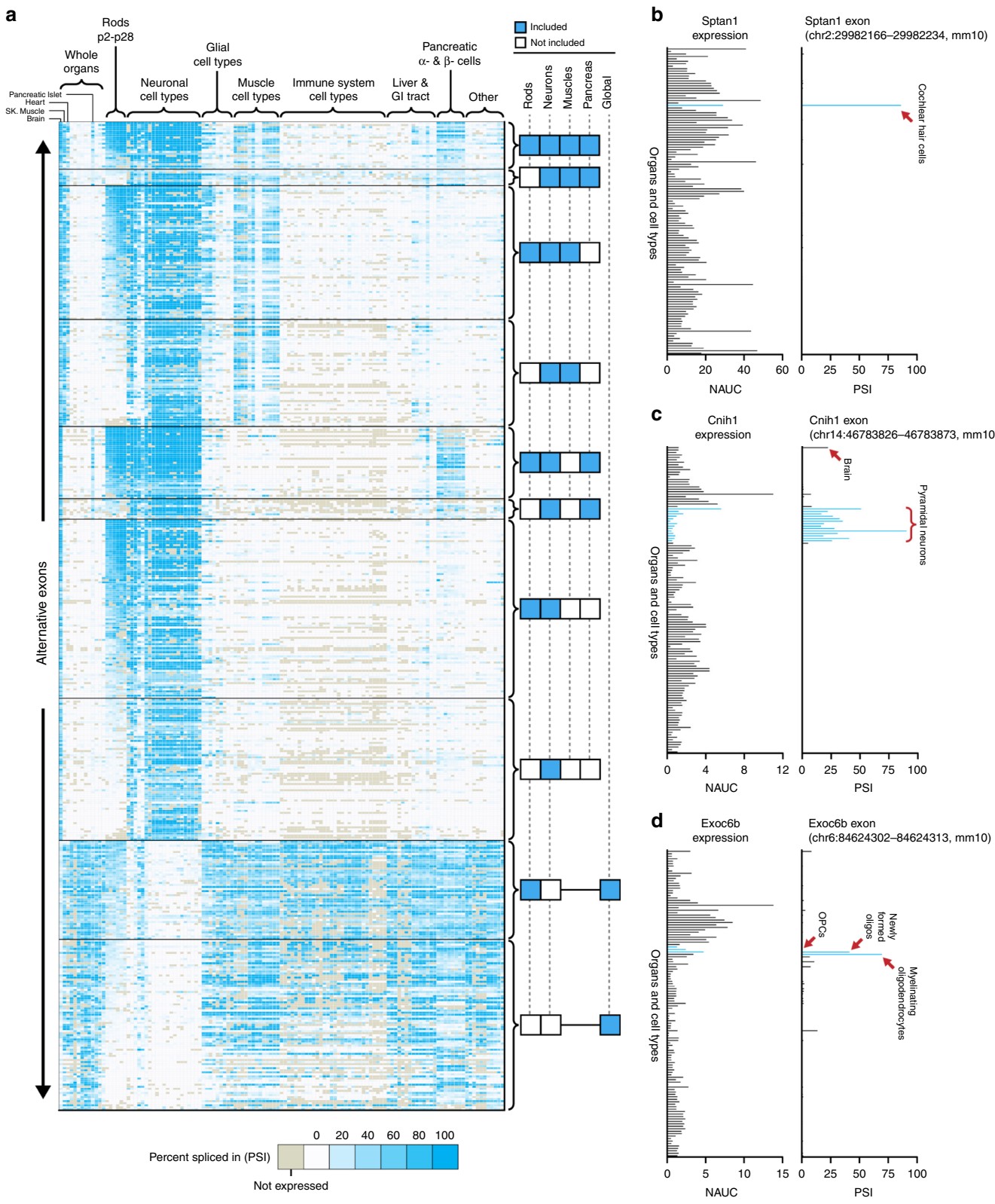

photoreceptor-specific exons revealed only 31 splicing events (found in 28 genes) that were common between both species. These 28 genes generally fall under pathways of cilia formation, neuronal connectivity and various metabolic pathways. Likewise, mutations in several genes are linked to retinitis pigmentosa or intellectual disability, underscoring their functional importance (Fig. 2c). Among the 31 rod-specific splicing events, 17/31 have

been previously identified while 14/31 have not been reported in the literature (Supplementary Fig. 5). Comparison of results between mouse and human is important since there can be significant variation in splicing specificity between species. For example, an exon in mouse *Cep290* is only utilized by photoreceptors, but the alternative exon in human *CEP290* is constitutively spliced across all human tissues.

**Fig. 1 Alternative exons enriched in the nervous system (MESA compilation). a** Mouse RNA-Seq data sets were manually curated from the SRA, covering a broad range of cell types and organs. Cell type data sets were generated from various independent labs using FACS or affinity isolation. To test our algorithm, we identified alternative exons that were differentially spliced between neuronal cell types and other cell types in the body and found that exons could be generally clustered by their inclusion or exclusion in rods, neurons, muscles, pancreas, or global non-neuronal (right columns). Each row is an individual exon, and exon utilization is measured by a percent spliced in (PSI) ratio as indicated by gradient legend (bottom). The overlap between neuronal exons and muscle cell types agrees with previous observations from our work[51] and others[58–60], suggesting that these exons are at least partially activated by *Ptbp1* downregulation. There is only partial overlap between rod exons and neuron-enriched exons, which is not unexpected since rods do not express many neuron-enriched splicing factors (Supplementary Fig. 3). **b–d** Our splicing analysis method reliably identifies alternative exons that are unique to specific cell types. For example, an exon in *Sptan1* is specifically enriched in cochlear hair cells, despite ubiquitous expression across all organs and cell types (**b**). Likewise, an exon in *Cnih1* is specifically enriched in pyramidal neurons (**c**) and an exon in *Exoc6b* is selectively enriched in myelinating oligodendrocytes (**d**).

**Cross-validation of splicing analysis using single-cell data**. To further test the sensitivity of our method, we incorporated single-cell RNA-Seq data sets generated using full-length library strategies (e.g. SmartSeq, Fluidigm) into ASCOT as a compilation called CellTower and analyzed the PSI tables for cell type-specific splicing patterns. Droplet-based strategies that sequence short sequences from the polyA tail (e.g. DropSeq, 10x Genomics) are useful for gene-level quantification, but are unable to capture most alternative splicing events. By contrast, single-cell protocols that capture sequences across the full transcript can analyze splicing, given sufficient read depth. The Allen Brain Institute recently generated extremely high coverage single-cell data sets from adult mouse primary visual cortex[33] and clustered cells into 49 types (19 glutamatergic, 23 GABAergic, and 7 non-neuronal). Our analysis identified many alternative exons that showed not only differential usage between glutamatergic, GABAergic, and non-neuronal cell types, but also high variation within each broad grouping (Supplementary Fig. 6a). We were also able to identify mutually exclusive exons that varied among cell types, including those previously analyzed[33] (Supplementary Fig. 6b). Having validated our approach using single-cell RNA-Seq data, we then analyzed a data set containing both retinal progenitor cells and immature postmitotic precursor cells[50] from embryonic days E14, E18 or P2 that were profiled using Smart-Seq. We found that rod-specific exons in *Atp1b2* and *Ttc8* were detectable at low levels in early photoreceptor precursors, but not in retinal progenitors or postmitotic precursors of other retinal cell types (Supplementary Fig. 6c). Lastly, we confirmed that cell type-specific alternative exons in *Sptan1*, *Cnih1*, and *Exoc6b* (Fig. 1b–d) exhibited the same specificity in CellTower (Supplementary Fig. 6d).

**Using gene expression to identify candidate splicing factors**. What are the splicing factors that mediate rod-specific splicing patterns identified in MESA (Fig. 3a)? Although rods do not express many of the RNA-binding proteins (RBPs) thought to be involved in regulating alternative splicing in neurons[35] (Supplementary Fig. 3), they do show similar relative expression levels of Polypyrimidine tract-binding protein 1 (*Ptbp1*) and its paralog *Ptbp2* (Fig. 3b). High levels of *Ptbp1* repress many exons, and downregulation of *Ptbp1* accompanied by an upregulation of *Ptbp2* is an important prerequisite for neuronal splicing[36,37,51–60]. We hypothesized that certain RBPs, acting as splicing enhancers, could be selectively expressed in rods to mediate rod-specific splicing. We defined a list of putative splicing factors by identifying genes with RNA-binding domains, as determined by RBPDB[61], in the InterPro database. Overlapping rod-enriched genes with putative splicing factors revealed two top candidates, Musashi RNA-binding protein 1 (*Msi1*) and Poly(rC)-binding protein 2 (*Pcbp2*), that were expressed at much higher levels in rods relative to other cell types across the body. This is consistent with previous work demonstrating that *Msi1* promotes photoreceptor-specific splicing[35], although no studies have yet shown if *Pcbp2* performs a

similar function. We also considered the possibility that knockdown of constitutive splicing factors could activate rod-specific exons. However, analysis of 1,159 data sets from the ENCODE shRNA-Seq project[31,32] did not reveal any shRNA knockdown that could activate rod-specific exons (Fig. 4a).

**MSI1 and PCBP2 induce rod-specific splicing in non-neurons**. To test whether *MSI1* and *PCBP2* overexpression was sufficient to activate rod-specific exons, we transfected these proteins into HepG2 cells, a liver cancer cell line used by the ENCODE project. Initially, we found that normal transfection of *MSI1* or *PCBP2* could not activate rod-specific exons (Supplementary Fig. 7). However, given the extremely high expression of these factors in mature rods, we hypothesized that the average expression levels achieved by transfection were not high enough to induce rod-specific splicing. We therefore used FACS to isolate the most strongly GFP-positive *MSI1/PCBP2*-transfected HepG2 cells (with or without simultaneous knockdown of *PTBP1*) to more accurately reflect the expression levels of these splicing factors seen in mature rods. These robustly transfected cells had significant activation of rod-specific exons (Fig. 4a, Supplementary Data 3). Specifically, *PCBP2* activated a single rod-specific exon in the monocarboxylate transporter Basigin (*BSG*) independent of *PTBP1* knockdown; *BSG* is necessary for photoreceptor survival[42–44]. By contrast, high expression of *MSI1* activated an extremely broad range of exons and appears to strongly synergize with *PTBP1* downregulation (Fig. 4a, Supplementary Fig. 8). Not only are high levels of *MSI1* capable of activating rod-specific exons in HepG2 cells, we also observed activation of neuronal/muscle enriched exons thought to be regulated by *PTBP1* knockdown. Indeed, exon activation by *MSI1* alone was stronger than the effect of knockdown of *PTBP1* alone, suggesting a more complex interaction between *MSI1* and *PTBP1* (Supplementary Fig. 8).

**High levels of MSI1 lead to splicing-in of cryptic exons**. Interestingly, we also identified human-specific exons that were activated by high levels of *MSI1*, many of which are not found in any other tissue (Fig. 4a, Supplementary Data 3). These cryptic exons are likely incidentally activated by the extremely high expression of *MSI1* in the most robustly transfected cells. This contrasts with previous work in which cryptic exons are activated as a result of knockdown of splicing factor repressors[51,57,62–64]. *MSI1* has been reported to selectively bind to RNA that contain multiple UAG sequences[35,65–71]. A motif analysis reveals that UAG clusters are significantly enriched at the proximal intron of the 5' splice site (Fig. 4b); this pattern was consistent for both alternative and cryptic exons. Of the 31 rod-specific exons common between mouse and human, the majority are flanked by binding motifs for both PTBP1 and MSI1 (Supplementary Fig. 9). Using previously published *Msi1* CLIP-Seq[72] data, we also identified several photoreceptor-specific exons with *Msi1* CLIP peaks,

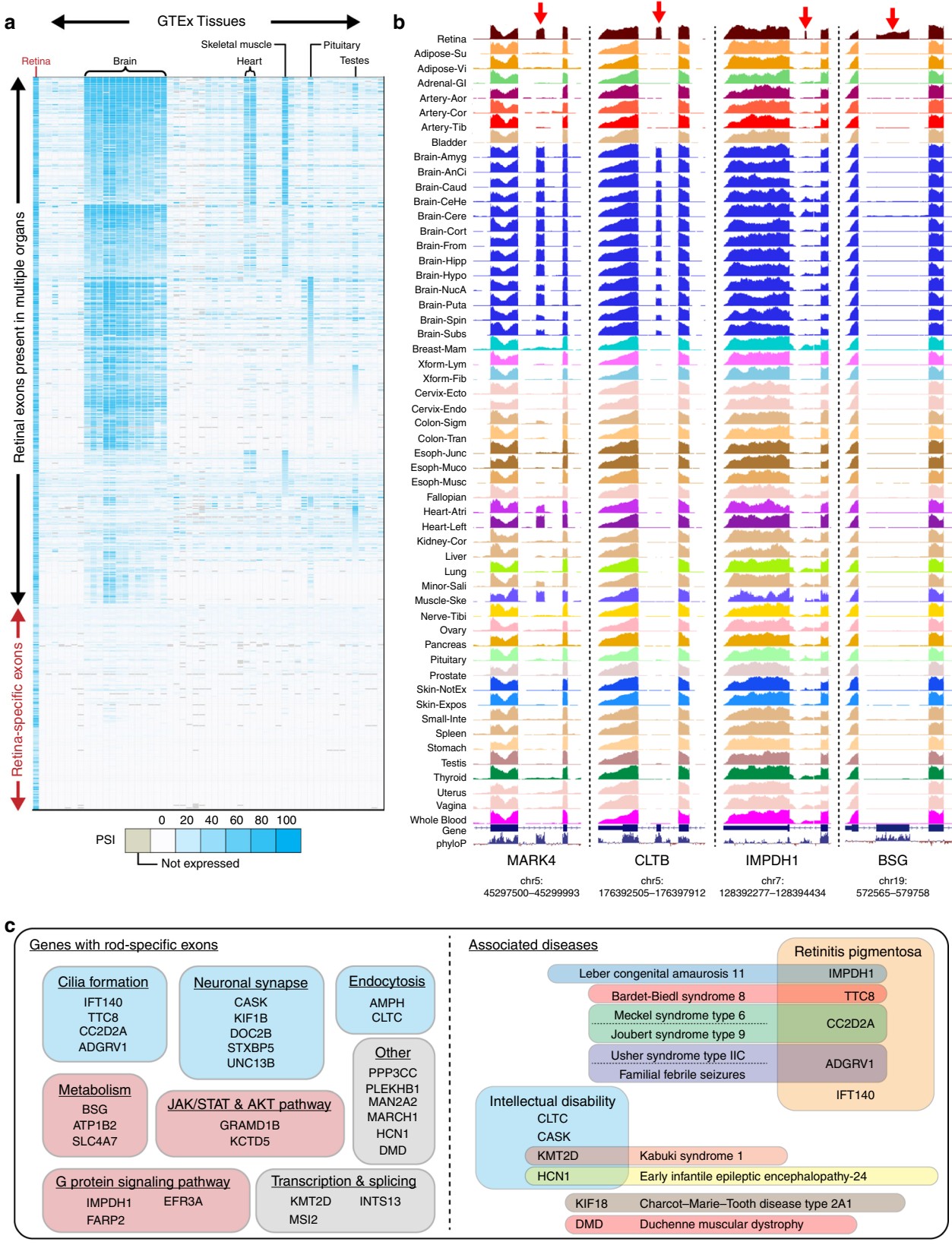

supporting a mechanism of direct interaction (Supplementary Fig. 10). UAG motif frequencies were compared to a baseline of all protein coding exons (<400 bp) in the GENCODE v28 basic gene annotation and exon examples are visualized in Fig. 5a.

**Msi1 knockdown abolishes photoreceptor-specific splicing.** Finally, we wanted to test whether loss of *Msi1* or *Pcbp2* function would result in a reduction of rod-specific exon splicing. We electroporated mouse retinal explants with shRNA or dominant

**Fig. 2 Splicing analysis of the GTEx database reveals retina-specific exons.** We cross-referenced our mouse splicing analysis with human RNA-Seq data sets generated by the GTEx consortium[30]. We also supplemented our analysis with data sets from peripheral retina[49], since GTEx did not sequence retinal tissue. **a** Similar to our mouse splicing analysis, we find many exons that are brain-enriched and that some of these exons are also present in skeletal muscle and heart. We also find that some exons are utilized in the pituitary and testes, tissues that are not present in our mouse database. We do not find overlaps in exon usage with GTEx pancreas because these data sets are generated from whole pancreas instead of pancreatic islets or α and β-cells. Importantly, we can identify alternative exons that are only spliced in retina. **b** Bigwig visualizations of raw GTEx RNA-Seq data. An exon in *MARK4* is present in retina, brain, and muscle. An exon in *CLTB* is present in retina and brain. Two exons, one in *IMPDH1* and one in *BSG*, are only present in retina and not in other GTEx tissues. **c** Comparing rod-specific exons in mouse with retina-specific exons in human yields a set of 31 exons that are likely important for photoreceptor function. Genes with rod-specific exons can be clustered under a variety of pathways from cilia formation and neuronal signaling to metabolism and GPCR pathways (left). Mutations in several of these genes are also associated with either retinitis pigmentosa or intellectual disability (right).

negative versions of *Msi1* and *Pcbp2* (Fig. 5b) and found that while reducing *PCBP2* function did not affect splicing of the rod-specific exon in *Bsg*, reducing *Msi1* function with shRNA or a dominant negative protein blocked rod-specific splicing (Supplementary Fig. 11). We confirmed that this result was specific to *Msi1* by electroporating shRNA targeting *Msi2* (an *Msi1* homolog), and found that *Msi2* shRNA did not reduce rod-specific splicing. We then analyzed the expression of a set of genes correlated with photoreceptor differentiation[73] and found that *Msi1* loss of function leads to expression patterns that resembled immature P2-P4 photoreceptors (Supplementary Fig. 11AA). Overall, expression of dominant negative *Msi1* mimics *Msi1* knockdown, but produces a somewhat weaker effect (Fig. 5b). Interestingly, while most rod-specific exons are reduced after *Msi1* knockdown, some rod-specific exons remain robustly incorporated (e.g. *Doc2b*, *Ppp3cc*, *Plekhb1*).

## Discussion

We have developed ASCOT, a resource that enables researchers to more easily perform cross-study splicing and expression analyses of public RNA-Seq data. ASCOT rapidly calculates exon PSIs and alternative splicing patterns using an annotation-free method that queries splice junction count tables. ASCOT's user interface and associated splicing/expression data sets are openly available at http://ascot.cs.jhu.edu. Although there have been past efforts to summarize public RNA-Seq data[28,29,74], ASCOT represents the largest effort to date to make alternative splicing and gene expression summaries of diverse data sets available to general researchers. ASCOT also demonstrates the value of using annotation-free methods to summarize publicly archived data.

Beyond scalability, ASCOT has several other advantages for analyzing cell type-specific alternative splicing. First, data set columns in splicing and expression summaries can be easily grouped and regrouped depending on the researchers needs, a feature that is especially useful for analyzing single-cell data (Supplementary Fig. 6). For example, clustering neonatal inner ear cells[75] based on primary cell type confirms that the exon in *Sptan1* (Fig. 1b) is only present in cochlear and vestibular hair cells and is absent in other inner ear cell types. Alternative splicing and gene expression data for these inner ear data sets, and a variety of other single-cell RNA-Seq studies, are available under the CellTower compilation of ASCOT (http://ascot.cs.jhu.edu). Data set clustering can also help identify alternative exons in bulk data that may be missed due to low gene expression. For example, by clustering GTEx data sets by organ, we can identify many exons that are differentially utilized between brain and heart that were not detected in Leafcutter's shiny app visualization, LeafViz (https://leafcutter.shinyapps.io/leafviz/)[11] (Supplementary Fig. 13). Second, ASCOT does not require transcript references to identify alternative splicing events, and is therefore unbiased toward annotated or unannotated exons. We estimate that ~40-60% of mouse and ~10-30% of human cassette exons identified by ASCOT are unannotated (Supplementary Fig. 12). Third, ASCOT can answer custom queries that

go beyond the data sets summarized in this study. For example, we queried rod-specific exons across 50,062 public data sets in the SRA (SRAv2 Snaptron compilation) to estimate the frequency of retinal data sets in the public archive (Supplementary Fig. 14). We found 37 data sets (0.07%) that had high PSI levels of rod-specific exons, and confirmed that these data sets were indeed from human retina. Finally, ASCOT can harmonize single- and paired-end RNA-Seq data of various read lengths. By starting from a splice junction count table, ASCOT can analyze alternative splicing across tens of thousands of archived RNA-Seq data sets without having to restart each analysis from raw fastq reads.

ASCOT is currently limited by its inability to detect complex alternative splicing events that other algorithms[11,13] can identify. We intentionally targeted binary splicing decisions as they have a straightforward biological interpretation and represent the majority of alternative splicing events. However, complex splicing can certainly be modeled with nested decision trees that would still be compatible with a junction-walking strategy. We believe that splice junction count tables contain enough information to build these splice models. Also, ASCOT does not attempt to model biases that can distort junction counts, such as GC content or secondary structure. We plan for future versions of ASCOT to model and mitigate these effects.

We used ASCOT to study tens of thousands of data sets from SRA, ENCODE, GTEx. Analyzing splicing factor gene expression across various mouse cell types allowed us to identify *MSI1* and *PCBP2* as candidates for inducing rod-specific splicing patterns, while the ENCODE shRNA-Seq data confirmed for us that knockdown of constitutive splicing factors could not activate rod-specific exons. Taken together, these observations led to the hypothesis that manipulating certain splicing factors could lead to rod-like splicing patterns. Only with this hypothesis in mind were we able to generate new data to conclude that robust overexpression of *PCBP2* and *MSI1* combined with *PTBP1* knockdown was able to activate rod-specific exons, even in a non-neuronal cell line such as HepG2. This study is emblematic of a larger shift toward using public data sets, often pre-summarized or indexed, to generate hypotheses and narrow the scientific question prior to designing experiments and generating new data. Resources such as ASCOT can save researchers much time and effort, as well as create new avenues of research for smaller labs with limited funding.

Together, our results suggest a model of photoreceptor splicing regulation (Fig. 5c) whereby *MSI1* and *PTBP1* downregulation interact synergistically. *MSI1* overexpression leads to the incorporation of *PTBP1*-repressed exons, while *PTBP1* downregulation increases *MSI1*'s ability to activate rod-specific exons (Supplementary Fig. 8). We have also identified that *PCBP2* is another regulator of photoreceptor-specific splicing. The rod-specific exon in *BSG* is essentially undetectable in all non-retinal tissues and *PCBP2* overexpression increases the exon PSI to ~8% (PSI in photoreceptors is >80%). However, mouse retina electroporation of shRNA and dominant negative constructs targeting *PCBP2* did not reduce levels of the rod-specific *BSG* exon, suggesting that

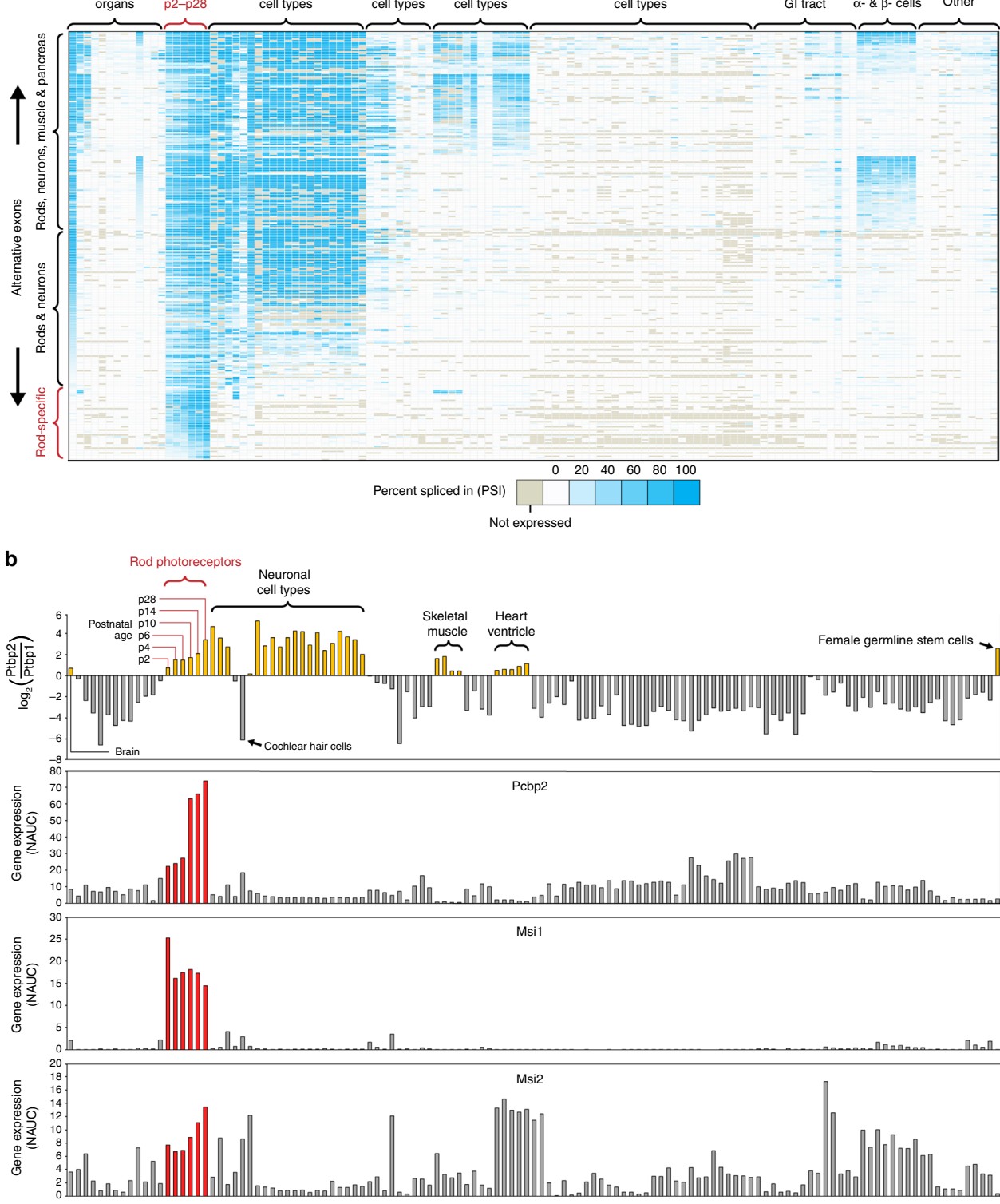

**Fig. 3 *Msi1* and *Pcbp2* are highly enriched in rod photoreceptors relative to other cell types (MESA compilation). a** Analysis of splicing patterns enriched in mouse rods reveals some exons that overlap with neurons, muscles, or pancreatic α and β-cells but also exons that are rod-specific and not present in other cell types. **b** Similarly to other neurons, rods downregulate *Ptbp1* and upregulate *Ptbp2*. The *Ptbp2/Ptbp1* ratio gradually increases in rods over development from P2 to P28. However, we also find that cochlear hair cell sensory neurons retain high levels of *Ptbp1*, suggesting that perhaps not all neurons switch from using *Ptbp1* to *Ptbp2*. To identify the factors that might mediate the splicing of rod-specific exons, we cross referenced genes with expression patterns that were enriched in rods with a list of putative RBPs and identified *Msi1* and *Pcbp2* as potential candidates. Note that the expression of *Msi2*, a paralog to *Msi1*, does not show the same selectively high expression in rods.

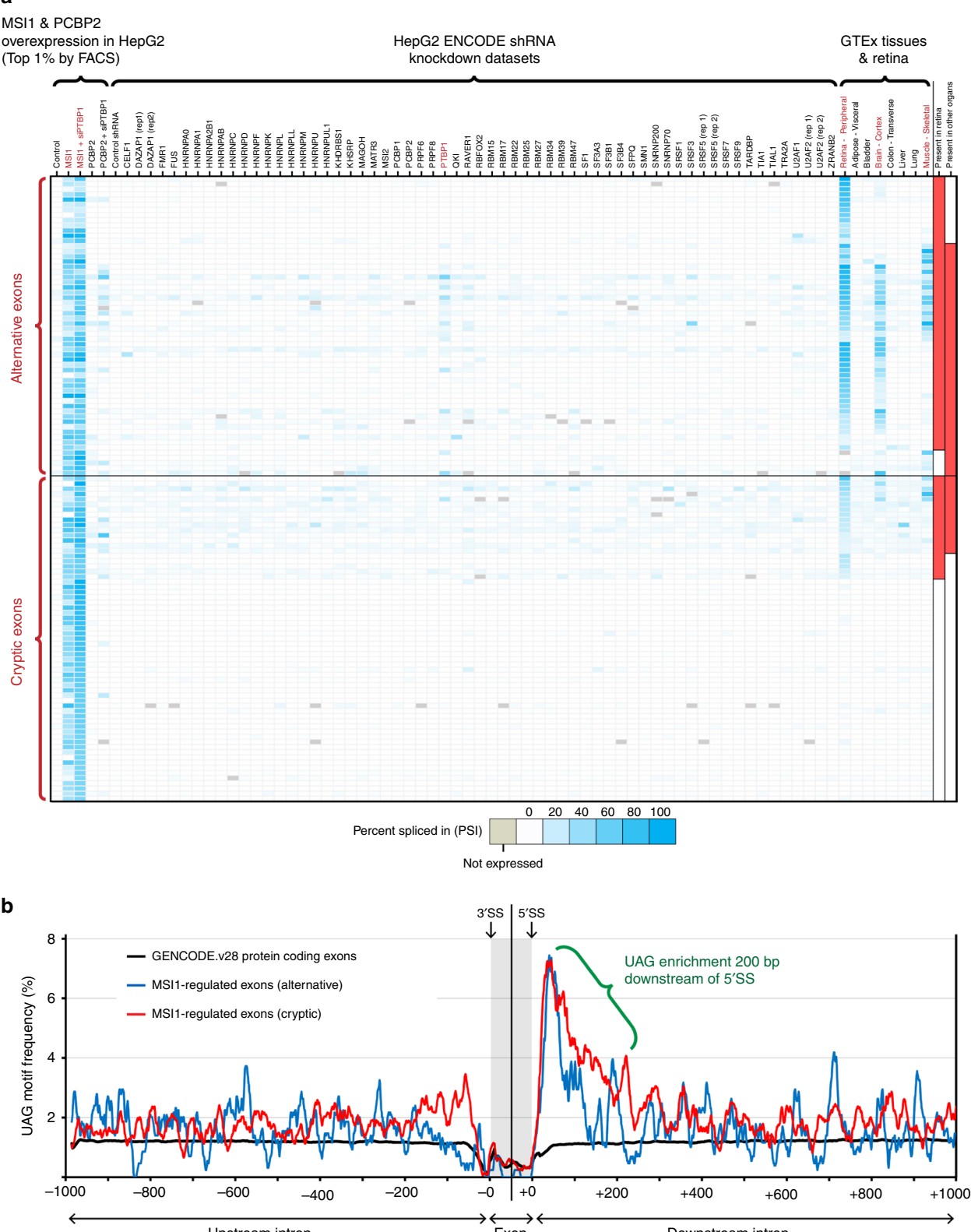

**b** UAG enrichment 200 bp downstream of 5′SS

*PCBP2* overexpression can activate the exon in non-neuronal cells but is not required to maintain splicing in mature photoreceptors. By contrast, knockdown of *Msi1* in electroporated mouse retina abolishes most of the rod-specific splicing events, leading to a delay in photoreceptor maturation (Supplementary Fig. 11).

Although high levels of *MSI1* are required for photoreceptor-specific splicing, our results indicate that *MSI1* expression levels must still be titrated, since excessive overexpression in HepG2 cells led to the incorporation of deleterious, cryptic exons (Fig. 4a).

**Fig. 4 Overexpression of *MSI1* and *PCBP2* in HepG2 cells activates rod-specific exons.** We analyzed human ENCODE shRNA-Seq data sets[31,32] to test whether knockdown of constitutive splicing factors activates rod-specific exons. We also sought to test whether overexpression of *MSI1* and *PCBP2* activates rod-specific exons in the liver cancer HepG2 cell line (used by the ENCODE consortium). **a** Each row represents alternative or cryptic exons. Columns represent RNA-Seq data sets, grouped into three categories: 1. HepG2 control and overexpression data sets generated in our lab, 2. HepG2 shRNA-Seq data sets generated by the ENCODE consortium (only a subset of the full data representing well known splicing factors, full database of all shRNA-Seq data sets is available at http://ascot.cs.jhu.edu/data), 3. Various GTEx tissues[30] and peripheral retina[49] as reference data (retina, brain, and skeletal muscle are highlighted in red). No HepG2 shRNA knockdown data sets are capable of activating rod-specific exons but overexpression of *MSI1* followed by FACS isolation of the top 1% expressing cells can successfully activate rod-specific exons. Unexpectedly, robust *MSI1* overexpression can also activate neuronal- and muscle-enriched exons. These patterns can also be modestly detected in the *PTBP1* shRNA knockdown ENCODE data set, but *MSI1* expression alone induces a greater level of activation. Furthermore, we overexpressed *MSI1* with *PTBP1* siRNA and found that this combination appeared to act synergistically in activating retina/brain/muscle exons and led to a general increase in rod-specific exon incorporation (Supplementary Fig. 8). Interestingly, we also find that *MSI1* overexpression can activate a set of cryptic exons that are not found in any other human tissue (**b**) Motif analysis of intronic sequences (±1000 bp from 5′ and 3′SS) and exonic sequences (±50 bp from 5′ and 3′SS) reveals that UAG motifs are enriched in a 200 bp intronic window proximal to the 5′SS. UAG motifs are the consensus binding site for *MSI1*[65,67,68]. Repetitive CU/UC elements, the binding site for *PTBP1*, can also be found upstream of most rod-specific exons (Supplementary Fig. 9).

These cryptic exons reinforce the importance of obtaining human RNA-Seq data at the resolution of individual cell types, as there can be significant differences in splicing between mouse and human. With ASCOT, we identified 31 photoreceptor-specific splicing events that are common between mouse and human. However, this analysis is incomplete since isolated mouse photoreceptors were compared to human retina as opposed to isolated human photoreceptors. More remains to be understood about splicing in the retina since neighboring cell types, epigenetic states, and/or developmental timing may play a role in mediating optimal photoreceptor splicing. Conditional knockout of *Msi1* in the adult retina will help clarify these results, as will single-cell sequencing of human retinal organoids.

ASCOT is part of a larger effort to make gene expression and alternative splicing data more accessible to the general researcher[7,8]. By reducing the initial barriers to data analysis, we hope to accelerate cross-disciplinary work and foster unexpected discoveries.

## Methods

ASCOT data tables, software, and interactive browser are available at http://ascot.cs.jhu.edu.

**Publications used as data sources and bigWig visualization on the UCSC Genome Browser**. All RNA-Seq data used for this study was obtained from various publication as documented on the ASCOT web resource (http://ascot.cs.jhu.edu/ds/ds_list.html). To visualize individual data sets, bigWigs were generated from aligned bam files and compiled as UCSC TrackHubs. Instructions for visualizing this data is linked on the ASCOT web resource (http://ascot.cs.jhu.edu/ucsctracks.html).

**ASCOT splicing analysis methodology and software**. A detailed description of ASCOT's splicing analysis methodology is available in Supplementary Fig. 1. Briefly, ASCOT uses an exon-centric approach to consider only the local regions of a splice graph and analyzes these elements independently from one another. We focus on four binary splicing decisions: cassette exons, alternative splice site exon groups that share the same exclusion junction, linked exons, and mutually exclusive pairs of exons. Our method for splicing analysis relies on evidence from RNA-Seq split-read alignments (i.e. splice junctions), as opposed to coverage. By grouping splice junctions based on shared start or end coordinates, closed loops can be identified where we can start from any coordinate and trace a path through an alternating series of exons and introns that leads back to original starting coordinate. For binary splicing events, there are will be two independent loops that share the same exclusion junction conditions. All scripts used to generate ASCOT are available on a GitHub repository at https://github.com/jpling/ascot.

**HepG2 cell culture, transfection, and FACS isolation**. HepG2 cells (ATCC, HB-8065) were cultured in Eagle's Minimum Essential Medium (Quality Biological, 112-018-101CS) supplemented with 1x GlutaMAX (ThermoFisher Scientific, 35050061), 10% FBS (Corning, 35-010-CV) and 1% Penicillin-Streptomycin (ThermoFisher Scientific, 15070063). siRNA targeting PTBP1 (Sigma, SASI_Hs01_00216644) or eGFP as negative control (ThermoFisher Scientific, AM4626) were transfected using Lipofectamine 3000 (Thermo Fisher Scientific,

L3000-008) following the manufacturer's protocol. For overexpression of MSI1 and PCBP2, Ultimate ORF expression clones from ThermoFisher Scientific (MSI1 - IOH41182, PCBP2 - IOH4487) were cloned into pCAGIG (Addgene, 11159) and again transfected using Lipofectamine 3000. For experiments involving a combination of plasmid overexpression and siRNA knockdown, plasmids were first transfected at 0 h, siRNA were transfected at 24 h, and cells were processed two days later at 72 h. For FACS isolation, cells were dissociated using TrypLE (ThermoFisher Scientific, 12604013) to form a single-cell suspension and sorted by GFP fluorescence on a BD FACSCalibur in the JHMI Ross Flow Cytometry Core Facility.

**RNA extraction, library preparation, and RNA sequencing**. RNA was extracted from cell culture samples using the Monarch Total RNA Miniprep Kit (New England BioLabs, T2010S). Total RNA for RNA-Seq was then processed using the TruSeq Stranded Total RNA Library Prep Kit (Illumina) to construct RNA-Seq libraries. Sample libraries were then sequenced on an Illumina NextSeq. Data was de-multiplexed and converted into fastq files. Fastq files were then processed by the Rail-RNA spliced alignment program and incorporated into a Snaptron compilation.

RT-PCR primers used for novel exon validation:

Kctd5-forward: CTCCATACGGCACAACCAGT, Kctd5-reverse: GTAGCACC AAGGACCCTGTC, Flna-forward: TCGTAGCCCCTACACTGTCA, Flna-reverse: TTACACGGCTCCTCACCCTTG, Flnb-forward: CCCATGTGGTCAAGGTCTCC, Flnb-reverse: GTTACACCAAGCTCTCCGCT, Itgb1-forward: GGCGTCTGTGC AGAGCATAA, Itgb1-reverse: CAGTTGTCACGGCACTCTTG, Ywhae-forward: ACAGCCTCGTGGCTTACAAA, Ywhae-reverse: ACATCCTGCAGCGCTTCT TT, Vcl-forward: TCTCCCCCATGGTGATGGAT, Vcl-reverse: TGAATAAGTGC CCGCTTGGT, Farp2-forward: GTGTCACAGGAGCCAGTCAT, Farp2-reverse: TCCTTTTCTAGCCGAGTGCTG, Cltc-forward: TGATCCCGAGCGAGTCAA GA, Cltc-reverse: ACCAGGTCATGGACAAAGTCA, Ptprf-forward: TTGTCAT CGCCATCCTCCTG, Ptprf-reverse: TCCTTCAGCCCGATTGACTG, Ank3-forward: CGAGAACGACACGAAGGGAA, Ank3-reverse: GGCAACGTGTAA GGGAGTGA, Ppp6r3-forward: GCGGCATGAAGGAAACACTC, Ppp6r3-reverse: TGCACTCTTTGCAAGCAGCAT. Large differences in RT-PCR product sizes were resolved on 2% agarose gels. To resolve small differences in RT-PCR product sizes (<30 bp), an Agilent Fragment Analyzer was used instead.

**Ex vivo mouse retina electroporation**. All experimental procedures were pre-approved by the Institutional Animal Care and Use Committee of the Johns Hopkins University School of Medicine. For ex vivo electroporation experiments, postnatal day (P)2 to P4 mouse retinas were dissected into DMEM/F12 with 10% FBS and electroporated with 100 µg total plasmid in 100 µl volume using a BTX ECM 830 Generator. Electroporation was performed using six square pulses of 50 volts and 50 milliseconds duration with a 950 milliseconds interval between pulses. Retinas were then cultured on 0.2 µm Whatman Nucleopore Track-Etched Membranes (MilliporeSigma, WHA110406). At P14, electroporated retinas (8–16 per condition) were dissociated into single-cell suspension using the Worthington Papain Dissociation System (Worthington, LK003150) and GFP-positive cells were isolated with FACS. For shRNA knockdown, we used pre-validated constructs from The RNAi Consortium to knockdown Msi1 (TRCN0000098550), Msi2 (TRCN0000071974), and Pcbp2 (TRCN0000120931) and control shRNA (MilliporeSigma, SHC005). Electroporated shRNA plasmids were mixed with the pCAGIG plasmid (Addgene, 11159) at a ratio of 3:1 by weight (shRNA:pCAGIG) to label electroporated cells with GFP. Dominant negative constructs were generated using an N-terminal truncated PCBP2 sequence (ΔKH1-PCBP2, aa125–365) and a C-terminal truncated MSI1 sequence (aa1-199). Sequences were cloned into pCAGIG. Empty pCAGIG vector was used as a second control.

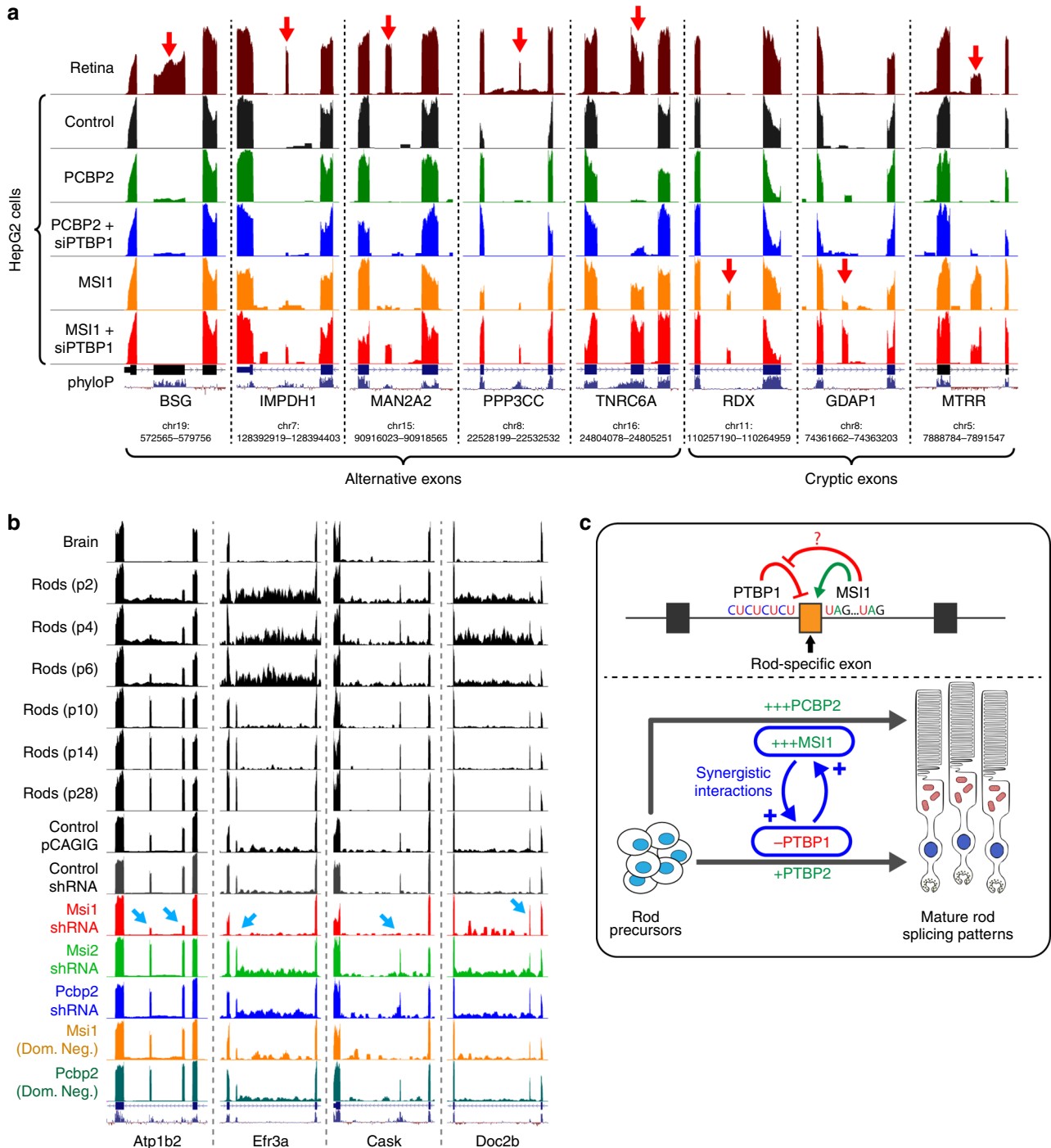

**Fig. 5 Visualization and pathway analysis of rod-specific exons and electroporation of mouse retina. a** UCSC visualization of rod-specific exons, both alternative (*BSG, IMPDH1, MAN2A2, PPP3CC, TNRC6A*) and cryptic (*RDX, GDAP1, MTRR*). Overexpression of *PCBP2* activates the rod-specific exon in *BSG*, independent of *PTBP1* knockdown. By contrast, *MSI1* overexpression in combination with *PTBP1* knockdown activates both rod-specific exons (*IMPDH1, MAN2A2, PPP3CC*) and pan-neuronal exons (*TNRC6A*). High levels of *MSI1* overexpression can also activate cryptic exons (*RDX, GDAP1*) that are not found elsewhere in the body. However, some of the cryptic exons activated by *MSI1* overexpression are not cryptic and can be found in photoreceptors (*MTRR*). **b** Electroporation of *Msi1* shRNA in mouse retina abolishes nearly all rod-specific splicing (*Atp1b2, Efr3a, Cask*). However, some rod-specific exons (*Doc2b*) remain unaffected (Supplementary Fig. 11). **c** Model for the regulation of photoreceptor-specific splicing. Increased expression of *MSI1* and downregulation of *PTBP1* act synergistically to activate rod-specific exons, while increased *PCBP2* expression can activate the rod-specific exon in *BSG*.

**Generation of Snaptron compilations**. Raw RNA-Seq fastq reads from all the input accessions were first analyzed using Rail-RNA, a cloud-enabled spliced alignment program that can analyze many samples at once[5,6]. Rail-RNA outputs a few summaries for each run accession, including a table of splice-junction evidence. In this table, each row is a splice junction and each column is an individual run accession. The elements of the table give the number of times a spliced alignment from an individual (column) spanned a junction (row). These summaries are then composed and indexed using Tabix and SQLite, and all the associated metadata for the run accessions are indexed using Lucene, to form a Snaptron compilation. A Snaptron compilation can be queried via command line or via RESTful API queries.

**Reporting summary**. Further information on research design is available in the Nature Research Reporting Summary linked to this article.

## Data availability

RNA-Seq data sets has been deposited in the NCBI Sequence Read Archive. The accession numbers for the sequencing data in this paper are SRP219036 (HepG2 overexpression) and SRP218930 (retina electroporation).

## Code availability

Gene expression summaries, alternative splicing summaries, and visualization tools are available at the ASCOT online resource: http://ascot.cs.jhu.edu. Source code and software are available under the following github repository: https://github.com/jpling/ascot.

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

## Acknowledgements
We thank X. Zhang and the Johns Hopkins Ross Flow Cytometry Core Facility. We also thank the Johns Hopkins Deep Sequencing & Microarray Core Facility for sequencing services. This work was supported by grants from the NIH (R01EY020560 to SB, K99EY027844 to BSC, R01GM118568 to BL, and R01GM121459), a postdoctoral fellowship from the Johns Hopkins Kavli Neuroscience Discovery Institute to J.P.L., and seed funding from The Institute for Data Intensive Engineering and Science (IDIES) at Johns Hopkins University to B.L.

## Author contributions
J.P.L., B.L. and S.B. conceived and planned the experiments. J.P.L., C.W., R.C., A.N. and B.L. developed the computational methodology and framework. J.P.L., P.J.L., D.G., C.P.S., B.P., L.J., B.S.C. and A.V. carried out in vitro experiments and generated biological data. J.P.L., B.L. and S.B. wrote the paper.

## Competing interests
The authors declare no competing interests.
