## [Peer Review File · Nature Communications]

Reviewers' comments:

Reviewer #1 (Remarks to the Author):

In this manuscript, the authors described a computational pipeline ASCOT to identify and quantify alternative splicing events. As an example, they identified rod-specific alternative exons and proposed MSI1 and PCBP2 as the candidate regulators.

There have been various existing methods (e.g., MISO, SpliceTrap, rMATS, MAJIQ, DARTS) to quantify alternative splicing events and to identify tissue or cell-type specific alternative splicing. Many of them are also exon-centric and utilize de novo split-reads to analyze unannotated alternative splicing. Without any method comparison, it's difficult to judge whether ASCOT outperforms existing methods.

The example of rod-specific alternative splicing sounds interesting. But the identification of splicing regulators lacks direct evidence.

Specific comments:

1. ASCOT relies on de novo junction read mapping as well as transcriptome assembly to discover new alternative exons. Such de novo junction/middle exon discoveries are prone to false positives. It will be essential to perform validation experiments to verify the cell type-specific alternative splicing events, since the authors claim that majority of the cell-type-specific exons are unannotated.
2. For PSI calculation, there is no description about how they combine the 3' PSI and the 5' PSI. What if the 3' PSI is very different from 5' PSI due to experimental bias? Is there any coverage threshold when identifying closed junction loops?
3. ASCOT excludes minor inclusion junction but includes minor exclusion junction in PSI calculation. Is there a specific reason for doing this? Since "This sum of minor exclusion junctions is important to ensure an accurate PSI calculation", what's the impact of minor inclusion junction for PSI calculation? ASCOT requires the primary inclusion junction representing >70% of all junction counts. Will the change of threshold affect the discovery?
4. In Figure 1A, what are the specific thresholds or methods to identify differential splicing between tissues/cell types? Will ASCOT treat PSI from high coverage and low coverage equally? How to define "not expressed" PSI? Does "not expressed" mean no junction reads or no exon body reads?
5. To prove that MSI1 and PCBP2 are regulators of rod-specific alternative splicing, experiments such as CLIP-seq are needed to show the direct binding. The overexpression experiments in a liver cancer cell line is not convincing enough, because the cellular environments are too different and alternative splicing network is highly coordinated. The no-signal results in ENCODE shRNA-seq for tumor cell lines also implies that cellular environments are important for alternative splicing regulation.
6. Signals from the knockdown experiments in retinal explants are weak, which further raises the concern of whether MSI1 and PCBP2 are truly corresponding regulators.
7. For the overexpression and knockdown experiments, besides the impact on rod-specific exons, do they also change the splicing of other exons?

Reviewer #2 (Remarks to the Author):

Ling et al. presented a computational method called ASCOT that aims at curating large-scale RNA-seq datasets from public data depositories and identifying alternative splicing variants across different tissue samples, cell types, etc. This method uses splice junction mapping reads combined with an exon-centric strategy to perform junction walk graphs, and is able to detect four subcategories of binary exon alternative splicing. The authors then applied this method across a large number of datasets covering human and mouse tissues/samples at both bulk and single cell level. They discovered cell-type specific alternative splicing of exons in rod photoreceptors in neuron subtypes, and also identified potential splicing regulators that contribute to the specific exon splicing pattern in the cell type.

The power of this method lies in its ability to process and scan many large datasets at the same time and build/profile a binary exon alternative splicing database from there. This feature will be needed by researchers especially now that vast number of RNA-seq datasets are generated every day while our ability to interpret them in an integrated and consistent manner is falling behind.

That said, when it comes to actual splicing analysis design and integrative statistical build, this method does not provide new insights nor development for the field. First of all, the concept of using splice junction reads for annotation-free style splicing analysis has been applied by many previous methods, such as rMATS (Shen et al. 2014), MAJIQ (Vaquero-Garcia et al. 2016), LeafCutter (Li et al. 2017), and JUM (Wang et al. 2018). Secondly, the concept of using "junction walk" (graph theory) to compile alternative splicing patterns has also been applied, for example rMATS (Shen et al. 2014), MAJIQ (Vaquero-Garcia et al. 2016) and JUM (Wang et al. 2018). In fact JUM is able to apply junction "walk" graphs to reconstitute all alternative splicing pattern categories (cassette exons, mutually exclusive exons, A5SS, A3SS, intron retention, complex splicing patterns) in an annotation-free manner, while the compilation of ASCOT here is restricted to only binary exon alternative splicing (and the authors should cite JUM in the manuscript).

In addition, after compiling the binary exon alternative splicing it seems that ASCOT will simply output a percent-splice-in (PSI) number for the specific exon in every sample analyzed, and then do clustering based on the PSI number distribution across different samples. There is no integrative statistical analysis involved, such as incorporating variance from biological replicates etc. which has been widely applied in many differential splicing analysis tools early on and proved to be very useful, such as DEXSeq (Anders et al. 2012) and all the previously mentioned software tools above. In this aspect ASCOT seems to be more like a tool for profiling and database generation for exon splicing.

In all, ASCOT has a unique advantage in scaling up alternative splicing analysis to vast number of RNA-seq datasets at the same time. It will be most useful in interpreting tissue- or cell type-specific exon alternative splicing for applications that profile through large-scale public datasets.

Below are two of my major comments:

- 1) ASCOT does have an annotation-free feature by utilizing splicing junction mapping reads. However, the follow-up exon-centric strategy requires an accurate annotation for exons, and hence is annotation-dependent. And the quality of the downstream alternative splicing analysis will depend on the accuracy of the exon annotations. It seems that ASCOT uses strategies similar to tools such as cufflinks that perform ab initio assembly of the transcriptome from RNA-seq data, and then extracts just local exon information from there. I am wondering if the authors can prove that the exon annotation is reliable from this approach, as usually the quality of ab initio transcriptome assembly from shot-gun sequencing reads is problematic.
- 2) Can the authors specify whether, and how do they handle variance from biological replicates, or variance from cells of the similar type when using a Snaptron Database for junction counts to profile tissue-specific exon alternative splicing?

References:

Anders et al. Detecting differential usage of exons from RNA-seq data. *Genome Res.* 22, 2008–17 (2012).

Shen et al. rMATS: robust and flexible detection of differential alternative splicing from replicate RNA-Seq data. *Proc. Natl. Acad. Sci. U. S. A.* 111, E5593-601 (2014)

Vaquero-Garcia et al. A new view of transcriptome complexity and regulation through the lens of local splicing variations. *Elife* 5, e11752 (2016).

Li et al. Annotation-free quantification of RNA splicing using LeafCutter. *Nat. Genet.* 50, 151–158 (2018)

Wang et al. JUM is a computational method for comprehensive annotation-free analysis of alternative pre-mRNA splicing patterns. *Proc. Natl. Acad. Sci. U. S. A.* 115(35):E8181-E8190 (2018)

Reviewers' comments:

Reviewer #1 (Remarks to the Author):

In this manuscript, the authors described a computational pipeline ASCOT to identify and quantify alternative splicing events. As an example, they identified rod-specific alternative exons and proposed MSI1 and PCBP2 as the candidate regulators.

There have been various existing methods (e.g., MISO, SpliceTrap, rMATS, MAJIQ, DARTS) to quantify alternative splicing events and to identify tissue or cell-type specific alternative splicing. Many of them are also exon-centric and utilize de novo split-reads to analyze unannotated alternative splicing. Without any method comparison, it's difficult to judge whether ASCOT outperforms exiting methods.

We agree with the reviewer that it is difficult to directly compare ASCOT to existing methods, since a comparison of these methods across thousands of different datasets generated by different labs is nontrivial. Our goal with ASCOT is not to outperform existing methods, but rather perform calculations that most methods are already able to do robustly (e.g. PSI calls for cassette exons) and apply them at a large scale. The methodology we have developed for ASCOT is scalable across the entire public sequence archive.

The example of rod-specific alternative splicing sounds interesting. But the identification of splicing regulators lacks direct evidence.

In Specific Comment #5, the reviewer suggests performing experiments like CLIP-seq to detect direct binding of Msi1 to rod-specific exons. However, CLIP-seq is extremely technically challenging to perform in the retina due to the limited amounts of tissue available. Although it is possible that a transgenic strategy using modified Msi1 could allow cross-linked pulldown of Msi1 targets in photoreceptors, no such genetic lines exist. Nevertheless, CLIP-seq studies of Msi1 and Pcbp2 have been performed *in vitro* with various cell lines. We demonstrate that although these non-retinal cell lines do not express all the genes present in the retina, we do in fact observe Msi1 CLIP-seq peaks adjacent to several photoreceptor-specific exons. This data has been incorporated into Supplemental Figure 10 in the manuscript, and is also included below as Figure 6 in this rebuttal letter.

Specific comments:

1. ASCOT relies on de novo junction read mapping as well as transcriptome assembly to discover new alternative exons. Such de novo junction/middle exon discoveries are prone to false positives. It will be essential to perform validation experiments to verify the cell type-specific alternative splicing events, since the authors claim that majority of the cell-type-specific exons are unannotated.

We thank the reviewer for this suggestion and have validated the existence of 11 newly identified alternative exons using RT-PCR. ASCOT has enabled the identification of many exons not annotated in GENCODE release M20 (e.g. Supplemental Excel File 1). We chose to validate a set of these exons that belong to genes widely expressed across the body. This validation has been added to the manuscript under Supplemental Figure 4 and included in this rebuttal letter as Figure 1 below. To perform this validation, we isolated RNA from 12 different tissues across the body and designed primers flanking each alternative exon. For alternative exons >50bp, we used agarose gel electrophoresis to resolve DNA bands. For exons <50bp, we used an Agilent Fragment Analyzer due to its ability to resolve fragments >5bp.

A

B

Figure 1. RTPCR validation of novel exons identified by ASCOT. Using the PSI data available through ASCOT, we identified unannotated cell type-specific exons that reside in ubiquitously expressed genes. We then designed primers spanning these alternative exons to yield a constitutive RT-PCR product and a larger RT-PCR product (green arrows) that contains the spliced-in cell specific exon. RT-PCR across a panel of tissues confirms the cell type-specificity predicted by ASCOT. **(A)** We used 2% agarose gels to resolve large size differences between RT-PCR products. **(B)** For smaller differences between RT-PCR products (<30bp), we used an Agilent Fragment Analyzer.

2. For PSI calculation, there is no description about how they combine the 3' PSI and the 5' PSI. What if the 3' PSI is very different from 5' PSI due to experimental bias? Is there any coverage threshold when identifying closed junction loops?

We thank the reviewer for identifying the missing information in the manuscript. A limited description of how 3' PSI and 5' PSI are combined is available in within the github repository (<https://github.com/jpling/ascot>), but no such description is listed in the Methods or Supplemental Figures. We have added a description of our 3' and 5' PSI averaging in the legend for Supplemental Figure 1.

In brief, we are well aware of variations between 3' and 5' PSI due to systematic underrepresentation of splice junctions. The source of biases in splice junction counts can be attributed to various technical confounders such as GC bias & random hexamer priming:

(Hansen KD et al., Nucleic Acids Res. 2010, PMID:20395217)

(Hansen KD et al., Biostatistics 2012, PMID:22285995)

(Love MI et al., Nat Biotechnol. 2016, PMID:27669167)

For example, one of the most prominent examples we have come across can be found in mouse pyruvate kinase m1/2 (Pkm) exon 9b (Figure 2A). In Pkm exon 9b, the 3' splice junction is drastically underrepresented as compared to the 5' splice junction and this underrepresentation is consistent across independently generated RNA-Seq samples. In contrast, no such variation is observed in any human RNA-Seq sample from GTEx (Figure 2B).

When we plot the difference between 3' and 5' PSI (Δ PSI) for all PSI events in ASCOT from lowest to highest, we observe a sigmoid distribution where the inflection points are at roughly 20 Δ PSI and 80 Δ PSI. Most exons detected by ASCOT have fairly consistent 3' PSI and 5' PSIs (83% of exons have a Δ PSI < 20). However, 11.7% of exons have a median Δ PSI > 80 and manual inspection confirms that these events are false positives or embedded splicing events (see Figure 4).

Initially, we considered an exclusion criteria of exons with a Δ PSI > 20. However, given that many important alternative exons such as Pkm exon 9b are above this threshold, we instead chose a more conservative threshold of Δ PSI > 80. We believe that this threshold excludes a reasonably large fraction of false positives, while maintaining the comprehensiveness of ASCOT. Furthermore, investigators who wish to adjust this threshold for higher or lower stringency can do so using the --f parameter in ascot_psi.py, as documented in the github repository.

Finally, we recognize that no solution can be perfect and that there will inevitably be false positives within a large database such as ASCOT. To this end, we and other groups are working on methods to normalize underrepresented splice junction counts, but such strategies will need to be accepted and standardized across the bioinformatics community before we move away from raw splice junction counts.

Figure 2. Pyruvate Kinase M1/2 (Pkm) contains a pair of mutually exclusive exons (9a and 9b) that are alternatively spliced across tissues. In **mouse** Pkm, the 3' splice junction of exon 9b appears to be underrepresented compared to the 5' splice junction (red arrow) and this bias is present in nearly all independently generated mouse RNA-Seq samples. However, in **human** PKM, no such underrepresentation is found (green arrow) and the 5' and 3' PSI of exon 9b are nearly equivalent.

Figure 3. Difference between 5'PSI and 3'PSI (Δ PSI) sorted from lowest to highest across all mouse exons identified in ASCOT. For most exons, Δ PSI is below 30, while false positives can be frequently found when Δ PSI is > 80 . By default, ASCOT uses a cutoff of Δ PSI > 80 , which removes $\sim 11.7\%$ of exons.

3. ASCOT excludes minor inclusion junction but includes minor exclusion junction in PSI calculation. Is there a specific reason for doing this? Since “This sum of minor exclusion junctions is important to ensure an accurate PSI calculation”, what’s the impact of minor inclusion junction for PSI calculation?

The primary reason for including “minor exclusion junctions” in the denominator of the PSI calculation is to prevent loss of information. In an ideal situation, minor exclusion junctions would not contribute significantly to the denominator of the PSI calculation. However, erroneous alignments and complex splicing events can be confounders, and this information would be lost if only the predicted inclusion/exclusion junctions were used to calculate PSIs.

For example, in Figure 4 below, the exon that spans coordinates B↔C is detected as a cassette exon comprised of inclusion junctions A↔B and C↔D as well as an exclusion junction of A↔D. However, the 5' PSI will be skewed due to the third splicing decision of A↔E. Such information would be lost if only A↔B and A↔D were used in the 5' PSI calculation. If a minor exclusion junction becomes a significant contribution to the PSI calculation, this can result in the exon being filtered out by the “primary inclusion junction threshold”, as described in the next section (Figure 5). Keeping minor exclusion junctions in the PSI calculation can therefore help reduce false positive rates.

Figure 4. Diagram illustrating an example of minor exclusion junctions confounding a PSI calculation.

ASCOT requires the primary inclusion junction representing >70% of all junction counts. Will the change of threshold affect the discovery?

Adjusting the primary inclusion junction fraction has a minimal effect on exon discovery, and this parameter is only used to reduce potential false positives (Figure 5, below). Only at stringent thresholds (primary inclusion junction >90% of all junctions) do we see significantly reduced exon discovery. We chose a threshold of 0.7 to reduce the number of false positives, while attempting to be as comprehensive as possible.

Figure 5. Effect of inclusion fraction threshold on exon discovery. A threshold of 70% only reduces exon discovery by 2.5%, but can eliminate many false positives.

4. In Figure 1A, what are the specific thresholds or methods to identify differential splicing between tissues/cell types? Will ASCOT treat PSI from high coverage and low coverage equally? How to define “not expressed” PSI? Does “not expressed” mean no junction reads or no exon body reads?

Yes, we want to treat gene expression and PSI calculations independently, as thresholding by expression would be subject to technical or biological confounders (sequencing depth, read length, sample preparation, library bias, etc). To achieve this, we simply set a threshold whereby PSI calculations must be supported by a minimum of 15 junctions per sample. Similar junction thresholds are used by other methods. For example, LeafCutter (PMID: 29229983) “iteratively removes introns that are supported by fewer than a specified number (default: 30) of reads across all samples”. Users can adjust this PSI junction threshold using the `--min` parameter in `ascot_psi.py`, as documented in the github repository.

5. To prove that MSI1 and PCBP2 are regulators of rod-specific alternative splicing, experiments such as CLIP-seq are needed to show the direct binding. The overexpression experiments in a liver cancer cell line is not convincing enough, because the cellular environments are too different and alternative splicing network is highly coordinated. The no-signal results in ENCODE shRNA-seq for tumor cell lines also implies that cellular environments are important for alternative splicing regulation.

Msi1 CLIP-Seq supports direct interaction with photoreceptor-specific exons. **(A)** Analysis of *PCBP2* eCLIP data from the ENCODE Project (Sloan et al. Nucleic Acids Res. 2016, PMID: 26527727) indicates that *PCBP2* does not directly interact with the photoreceptor-specific exon in *BSG*. However, *PCBP2* interacts strongly with the 3'UTR of *BSG*, possibly indicating an indirect mechanism of action. **(B-F)** Using previously published *Msi1* CLIP-Seq data generated in intestinal epithelium (Li et al. Cell Reports 2015, PMID: 26673327), we identify several exons with *Msi1* CLIP peaks. Given that intestinal epithelium do not splice-in photoreceptor-specific exons and do not express many photoreceptor-specific genes, the presence of *Msi1* CLIP peaks near photoreceptor-specific exons is significant. These intronic sequences would be spliced out and degraded, and should therefore be underrepresented in the CLIP data.

Figure 6. *Msi1* CLIP-Seq supports direct interaction with photoreceptor-specific exons. (A) Analysis of *PCBP2* eCLIP data from the ENCODE Project (Sloan et al. Nucleic Acids Res. 2016, PMID: 26527727) indicates that *PCBP2* does not directly interact with the photoreceptor-specific exon in *BSG*. However, *PCBP2* interacts strongly with the 3'UTR of *BSG*, possibly indicating an indirect mechanism of action. (B-F) Using previously published *Msi1* CLIP-Seq data generated in intestinal epithelium (Li et al. Cell Reports 2015, PMID: 26673327), we identify several exons with *Msi1* CLIP peaks. Given that intestinal epithelium does not splice-in photoreceptor-specific exons and do not express many photoreceptor-specific genes, the presence of *Msi1* CLIP peaks near photoreceptor-specific exons is significant. These intronic sequences would be spliced out degraded and should therefore be underrepresented in the CLIP data.

6. Signals from the knockdown experiments in retinal explants are weak, which further raises the concern of whether MSI1 and PCBP2 are truly corresponding regulators.

We are uncertain why the reviewer believes that the signals from the knockdown experiments in retinal explants are weak. For example, Figure 7 (right) shows that the photoreceptor-specific exon in *Kctd5* is entirely absent in photoreceptors when *Msi1* is knocked down with shRNA or disrupted with a dominant negative protein (green arrows). Notably, this *Kctd5* exon is one of the targets that we validated by RT-PCR in Figure 1.

Many other examples demonstrating loss of photoreceptor-specific splicing after *Msi1* knockdown are presented in Supplemental Figure 11 and Figure 5B of the main text.

7. For the overexpression and knockdown experiments, besides the impact on rod-specific exons, **do they also change the splicing of other exons?**

We have characterized the most significant alternative splicing changes upon overexpression of *Msi1* in Supplemental Excel File 3. There are significant changes in splicing when *Msi1* overexpression is combined with *Ptbp1* knockdown, as well as *Ptbp1* knockdown alone. However, *Ptbp1* is a well-known regulator of neuronal splicing and the effects of *Ptbp1* knockdown on splicing have been extensively reported in the literature:

Gueroussov et al. 2015, PMID: 26293963
Keppetipola et al. 2012, PMID: 22655688
Keppetipola et al. 2016, PMID: 27288314
Licatalosi et al. 2012, PMID: 22802532
Linares et al. 2015, PMID: 26705333
Llorian et al. 2016, PMID: 27317697
C. K. Vuong et al. 2016, PMID: 27094079
Zheng et al. 2012, PMID: 22246437

Figure 7. The photoreceptor-specific exon in *Kctd5* (validated by RT-PCR in Figure 1) is absent in photoreceptors when *Msi1* is knocked down with shRNA or disrupted with a dominant negative protein (green arrows). Other similar examples are presented in Figure 5B of the main text and Supplemental Figure 11.

Reviewer #2 (Remarks to the Author):

Ling et al. presented a computational method called ASCOT that aims at curating large-scale RNA-seq datasets from public data depositories and identifying alternative splicing variants across different tissue samples, cell types, etc. This method uses splice junction mapping reads combined with an exon-centric strategy to perform junction walk graphs, and is able to detect four subcategories of binary exon alternative splicing. The authors then applied this method across a large number of datasets covering human and mouse tissues/samples at both bulk and single cell level. They discovered cell-type specific alternative splicing of exons in rod photoreceptors in neuron subtypes, and also identified potential splicing regulators that contribute to the specific exon splicing pattern in the cell type.

The power of this method lies in its ability to process and scan many large datasets at the same time and build/profile a binary exon alternative splicing database from there. This feature will be needed by researchers especially now that vast number of RNA-seq datasets are generated every day while our ability to interpret them in an integrated and consistent manner is falling behind.

That said, when it comes to actual splicing analysis design and integrative statistical build, this method does not provide new insights nor development for the field. First of all, the concept of using splice junction reads for annotation-free style splicing analysis has been applied by many previous methods, such as rMATS (Shen et al. 2014), MAJIQ (Vaquero-Garcia et al. 2016), LeafCutter (Li et al. 2017), and JUM (Wang et al. 2018). Secondly, the concept of using “junction walk” (graph theory) to compile alternative splicing patterns has also been applied, for example rMATS (Shen et al. 2014), MAJIQ (Vaquero-Garcia et al. 2016) and JUM (Wang et al. 2018). In fact JUM is able to apply junction “walk” graphs to reconstitute all alternative splicing pattern categories (cassette exons, mutually exclusive exons, A5SS, A3SS, intron retention, complex splicing patterns) in an annotation-free manner, while the compilation of ASCOT here is restricted to only binary exon alternative splicing (and the authors should cite JUM in the manuscript).

In addition, after compiling the binary exon alternative splicing it seems that ASCOT will simply output a percent-splice-in (PSI) number for the specific exon in every sample analyzed, and then do clustering based on the PSI number distribution across different samples. There is no integrative statistical analysis involved, such as incorporating variance from biological replicates etc. which has been widely applied in many differential splicing analysis tools early on and proved to be very useful, such as DEXSeq (Anders et al. 2012) and all the previously mentioned software tools above. In this aspect ASCOT seems to be more like a tool for profiling and database generation for exon splicing.

In all, ASCOT has a unique advantage in scaling up alternative splicing analysis to vast number of RNA-seq datasets at the same time. It will be most useful in interpreting tissue- or cell type-specific exon alternative splicing for applications that profile through large-scale public datasets.

We appreciate the reviewer's thoughtful comments and agree with their assessment of ASCOT's strengths and weaknesses. We have also added citations for JUM and other mentioned references to the main text.

Below are two of my major comments:

1) ASCOT does have an annotation-free feature by utilizing splicing junction mapping reads. However, the follow-up exon-centric strategy requires an accurate annotation for exons, and hence is annotation-dependent. And the quality of the downstream alternative splicing analysis will depend on the accuracy of the exon annotations. It seems that ASCOT uses strategies similar to tools such as cufflinks that perform ab initio assembly of the transcriptome from RNA-seq data, and then extracts just local exon information from there. I am wondering if the authors can prove that the exon annotation is reliable from this approach, as usually the quality of ab initio transcriptome assembly from shot-gun sequencing reads is problematic.

We agree that transcript assemblers can run into trouble when trying to assemble and quantify full-length transcripts from short reads. We should clarify that we are using transcriptome assembly only to determine an inclusive set of cassette exons that we then use to query the Snaptron resource to form PSI tables. We are using the StringTie software in particular, which exhibits very high exon-level recall relative to other assemblers and which utilizes spliced alignments to a reference genome. This is in contrast to truly “ab initio” assemblers that assemble transcripts from scratch, without the benefit of a reference genome.

We do not assume the full-length transcripts produced by StringTie are accurate. Rather, we depend on StringTie’s cassette-exon recall, which others have shown to be high relative to other tools (PMID: 30052957). Nor do we depend on StringTie’s quantifications; our PSI estimates are calculated with data returned from Snaptron queries (which in turn are based on annotation-agnostic spliced alignment), not from StringTie’s estimates.

We further note that, while this is not how we created the ASCOT resource, an alternative approach would be to include all annotated cassette exons in the set of exons used to query Snaptron. In other words, there is nothing inherent in our approach that requires accurate transcriptome assemblies in order to form useful PSI summaries. We chose to start with assembled exons for ASCOT because a chief goal was to discover cell type-specific splicing patterns that were potentially novel and unannotated.

2) Can the authors specify whether, and how do they handle variance from biological replicates, or variance from cells of the similar type when using a Snaptron Database for junction counts to profile tissue-specific exon alternative splicing?

Unfortunately, we do not attempt to control for variance across biological or technical replicates as these parameters vary significantly across studies. We also find that many well cited studies, have only single replicate datasets, while single-cell studies have highly variable numbers (10’s-100’s) of replicates per annotated cell type. Ultimately, ASCOT relies on end users to properly consider any confounders that may be present after dataset merging. We plan to add user flexibility in selecting datasets from the public archive to enable custom analyses, but these plans will be implemented in the next version of ASCOT/Snaptron.

REVIEWERS' COMMENTS:

Reviewer #1 (Remarks to the Author):

This is a revised manuscript. The authors have addressed the concerns satisfactorily. I have no other questions.

Reviewer #2 (Remarks to the Author):

The authors answered the questions that I raised previously. ASCOT has its own weakness as a splicing analysis software tool (no variance from biological replicates incorporated, focus only on limited selections of AS patterns, no new/advancing development in methodology/design compared to other already existing tools etc.). On the other hand, it can be used like a splicing "annotation" or "classification" tool to incorporate large scale public datasets, which is indeed in need.

No further issues were raised by reviewers

REVIEWERS' COMMENTS:

Reviewer #1 (Remarks to the Author):

This is a revised manuscript. The authors have addressed the concerns satisfactorily. I have no other questions.

Reviewer #2 (Remarks to the Author):

The authors answered the questions that I raised previously. ASCOT has its own weakness as a splicing analysis software tool (no variance from biological replicates incorporated, focus only on limited selections of AS patterns, no new/advancing development in methodology/design compared to other already existing tools etc.). On the other hand, it can be used like a splicing "annotation" or "classification" tool to incorporate large scale public datasets, which is indeed in need.